# Study protocol of guided mobile-based perinatal mindfulness intervention (GMBPMI) - a randomized controlled trial

Siu-man Ng[1,2], Ling Li Leng[3]*, Ka Po Chan[4], Hay-ming Herman Lo[5], Albert Yeung[6], Shuang Lu[1], Amenda Wang[1], Hui Yun Li[1]*

1 Department of Social Work and Social Administration, The University of Hong Kong, Pokfulam, Hong Kong SAR, 2 Centre on Behavioural Health, The University of Hong Kong, Pokfulam, Hong Kong SAR, 3 The Department of Sociology, Zhejiang University, Zhejiang, China, 4 Centre of Buddhist Studies, The University of Hong Kong, Pokfulam, Hong Kong SAR, 5 Department of Applied Social Sciences, Hong Kong Polytechnic University, Pokfulam, Hong Kong SAR, 6 Department of Psychiatry, Harvard Medical School, Boston, Massachusetts, United States of America

* erinhy@connect.hku.hk (HYL); linglileng@zju.edu.cn (LLL)

**Data Availability Statement:** No datasets were generated or analysed during the current study. All relevant data from this study will be made available upon study completion.

## Abstract

### Background

Psychological distress is a common occurrence among women during the perinatal period. Maternal psychological distress (MPS) can also have a negative influence on neonatal outcomes such as infant health, child development or mother-child interaction. Hence, interventions to improve mental wellbeing during this period are vital. Mindfulness based intervention (MBI) has been found to be effective in reducing psychological distress. Delivery of MBI via the internet, making it accessible and inexpensive, is showing a promising positive effect in reducing psychological distress. A randomized control trial with sufficient power is required to confirm its positive effect among pregnant women. The positive effects of MBI have been found to be associated with heart rate variability (HRV) biofeedback; however, the efficacy of MBI on HRV has been rarely studied among pregnant women. Also, the potential association of HRV with MBI and psychological wellbeing needs further examination. This research aims to test the effectiveness of guided mobile-based perinatal mindfulness intervention (GMBPMI) among pregnant women experiencing psychological distress during the pre- and post-natal period, as well as examining the efficacy of GMBPMI on HRV.

### Method

This study is a randomized controlled trial that follows a parallel design. Consenting pregnant women in their second trimester (between 12th and 20th week gestation) will be randomly assigned to an intervention group (GMBPMI) or a control group (psychoeducation). The intended sample size is 198, with 99 participants in each group. Three levels of outcomes will be measured at baseline, post intervention in both the intervention and control groups, and at 36-week gestation and five-week postpartum. The primary outcomes include maternal psychological stress, mindfulness and positive appraisal HRV. Secondary outcomes are psychological and physical wellbeing. Tertiary outcomes include obstetric and

**Funding:** This research received General Research Fund funding scheme (17603520) from Research Grants Council, Hong Kong SAR. The funder provided support in the form of salaries for author [AW] and research materials, but did not have any additional role in the study design, data collection and analysis, decision to publish, or preparation of the manuscript. The specific roles of these authors are articulated in the 'author contributions' section.

**Competing interests:** The authors have declared that no competing interests exist.

neonatal outcomes, and social support. Analyses will follow an intention-to-treat method and repeated measures MANOVA will be conducted to compare changes in primary and secondary outcomes. A series of mixed-effects models will be fitted to assess the mediation effects.

## Discussion

This trial expects to increase understanding of GMBPMI on HRV and psychological wellbeing for pregnant women, with extended support in both pre-and post-natal periods. The study could also potentially provide evidence for delivery of cost-effective and accessible services to pregnant women.

## Trial registration

ClinicalTrials.gov: NCT04876014, registered on 30 March 2021. Protocol Version 1.0., 10 May 2021.

## Introduction

Pregnancy, giving birth and becoming a parent are three major life events that happen together. Were it not for the exhilaration of expecting a new life, many women would experience maternal psychological distress due to the naturally occurring physical, emotional and interpersonal changes [1]. MPS refers to the subjective feelings of distress or strain experienced by women during pregnancy [2]. The most common stresses faced by pregnant women include physical health concerns and psychological stress. Physically, they face increased medical and health concerns and physical discomfort during pregnancy. Psychologically, women experience difficulties in navigating a new identity, childbirth-related fear and anxiety, and the pressure of being a 'perfect mother' [3–5]. Research has identified many adverse consequences of maternal psychological distress (MPS) related to both mother and child. Some evidence suggests that MPS may be associated with poor obstetric and neonatal outcomes [6–8] including increased maternal obstetric complications such as analgesic use and unplanned caesarean delivery [9, 10]. Furthermore, fetal exposure to MPS can increase the risk of poor neonatal outcomes, including preterm delivery [11], low birth weight [4], low Apgar scores, smaller head circumference and major congenital anomalies [12, 13]. MPS not only affects perinatal maternal and infant health but may also result in poor postpartum mother-child interaction and child development [7, 14]. Recent evidence has also indicated that as well as depression, anxiety during pregnancy is related to developmental and behavioural issues in children [15].

Women in Hong Kong experiences MPS during perinatal period. Mindfulness intervention is increasingly being used to treat symptoms of anxiety, stress and depression [16, 17]. Accumulating evidence has suggested that mindfulness-based intervention (MBI) may be suitable and beneficial for women in coping with physical discomfort and negative emotions or thoughts during the perinatal period [18, 19]. Through mindfulness training, people can increase self-awareness of their emotions, thoughts and sensations, and cultivate curiosity and openness in their experiences without losing themselves in thoughts and feelings. Such skills are considered vital to pregnant women who experience physical discomfort intertwined with their emotions during the perinatal period.

In recent decades, empirical research has suggested the efficacy of MBI in reducing MPS. For example, a meta-analysis of 17 studies of prenatal MBI demonstrated significant pre- and post-improvements of medium effect size in reducing stress and depression [20, 21]. However, there were no significant postnatal improvements in stress and depression or in mindfulness skills for MBI participants versus the control participants [20]. Lau and colleagues [22] suggest a need to implement the intervention in the second trimester and extend the intervention to the third trimester and postnatal to prevent postpartum depression. MBI was also found to be associated with enhanced biological and infant outcomes. Higher maternal mindfulness of women at 22-week gestation significantly predicted a normal, as opposed to low, neonatal birth weight [23]. Van den Heuvel and colleagues [24, 25] showed that maternal mindfulness was negatively correlated with maternal anxiety during pregnancy, and positively associated with fewer infant self-regulation problems, a less difficult temperament, and greater control in infants at ten months old. In a randomized controlled trial examining the efficacy of MBI among pregnant women, Chan [26, 27] found mothers in the MBI group to have lower levels of the "stress hormone" salivary cortisol in the evening, and the infants at five months old had easier temperaments as well as improved positive appraisal. However, the significant effects were only observed among the 56% of participants who practiced mindfulness regularly [28]. Lack of continued motivation was the most common reason reported for giving up the practice, suggesting a need to extend the current MBI from the second trimester of pregnancy to the third trimester and postnatal period.

Previous studies have shown that MBI has led to increased heart rate variability (HRV) as well as decreased blood pressure and cortisol levels [29, 30], suggesting that MBI may have beneficial physiological effects for pregnant women [31]. Most recently, Braeken and colleagues [31] examined 156 pregnant women with self-reported mindfulness, psychological distress and HRV measures. The research highlighted an association between mindfulness and autonomic nerve system (ANS) changes, suggesting that mindfulness improves HRV measures and leads to better social and emotional development in both mother and offspring. However, few randomized controlled trials have been conducted to test the efficacy of MBI on ANS changes in pregnant women.

The present study aims to rigorously evaluate the effectiveness of a guided mobile-based perinatal mindfulness intervention (GMPMI). Specifically, we aim:

i.  to evaluate the effectiveness of GMPMI on psychological distress of women during the perinatal period;

ii.  to evaluate the impact of GMPMI on obstetric and neonatal outcomes;

iii.  to investigate HRV changes associated with GMPMI and its mediation effect between maternal psychological distress and neonatal outcomes.

## Method

### Design

The research adopts a parallel-armed, randomized controlled trial (RCT) design. Pregnant women in their second trimester will be recruited through Qualtrics (https://hku.au1.qualtrics.com/jfe/form/SV_2ofenm3v2OeCyJE). Eligible participants will be randomized to receive either the guided mobile-based perinatal mindfulness intervention (GMPMI) or the web-based perinatal psychoeducation program (WPPP). Repeated measures will be taken at four time-points: pre-intervention (T0); on completion of eight weekly PMI in the intervention group or eight weekly PPP on perinatal care in the control group (T1); 36-week gestation (T2); and five-week postpartum (T3).

## Setting

This research is based in Hong Kong. Adopting an online intervention mode, the location of the participants will not compromise their eligibility. The study sites will be the current living environment the participants. The participants will receive perinatal psychoeducation and support, which will be deliberated in the later part.

## Participants

Adult (age 18 or above) pregnant Chinese women in their second trimester (between 12th and 28th week gestation) will be invited to the study. This is when pregnancy becomes stable. Participants will be recruited from the community through advertisements posted on relevant websites, maternal discussion forums, Facebook and Twitter. The exclusion criteria are: 1) not able to understand Chinese (the intervention will be delivered in Chinese); 2) high-risk pregnancy status (e.g., preterm labor, placental abnormality, multiple gestations, required bed rest, or morbid obesity); and 3) current psychiatric disorders necessitating priority attention (e.g., schizoaffective disorder, bipolar disorder or current psychosis; organic mental disorder or pervasive developmental delay; current substance abuse or dependence; imminent suicide or homicide risk).

A screening interview via telephone or audio-chat will be conducted with each potential participant to assess eligibility as well as to explain the study in detail and to address concerns. The eligible participants will receive consent forms by email and will be asked to return these to the researchers.

## Sample size calculation

The previous eastern-based meditation intervention (EBMI) study reported an effect size of 0.44 in reducing prenatal distress and an attrition rate of around 20% [28]. In the proposed study we assume an overall moderate effect size of 0.4 and a conservative attrition rate of 25%. Setting a power at 90%, a significance level of $p < .008$ (corrected for six primary outcomes), a sample size of 99 per arm is needed for a two-arm repeated measures design (GPower, version 3.1). Thus, the targeted total sample size is 198. Considering this is a web-based intervention, we will consider expanding the sample size to ensure enough statistical power after preliminary data analysis.

## Conceptual framework

The study adopts the transactional theory of stress and coping as its underpinning theoretical framework. Transactional theory proposes how people react to stress, and the intensity of the reaction is heavily influenced by the mediating role of appraisal, the cognitive process through which meaning is ascribed to events [32, 33]. By reordering life priorities based upon one's values and goals and ascribing positive meaning to stressful events, meaning-focused coping helps to restore the resources that influence cognitive appraisals, sustain coping efforts over time and to provide relief from distress [34, 35].

Through continued mindfulness practice, we hypothesize that perinatal women can increase their maternal mindfulness. Increased maternal mindfulness can expand pregnant women's perceptual awareness and facilitate positive appraisal during the stressful perinatal period, thereby helping to reduce perinatal psychological stress. This is consistent with the results of a previous study, where positive appraisal was found to be significantly increased in a mindfulness group [28]. Moreover, by reducing perinatal psychological stress, the increase of maternal mindfulness is theorized to lead to higher HRV, suggesting increased

parasympathetic control. A decrease in perinatal psychological stress can bring about better birth outcomes (obstetric and neonatal) through increased parasympathetic activity. Fig 1 depicts this conceptual model.

## Four hypotheses

1. The intervention group would show a greater increase in maternal mindfulness and positive appraisal than the control group at T1, T2 and T3.

2. The intervention group would show a greater reduction in stress and depression than the control group at T1, T2 and T3 and these outcomes would be mediated by the enhancement in positive appraisal.

3. The intervention group would show a greater increase in HRV than the control group at T1, T2 and T3 and these outcomes would be mediated by the reduction in stress and depression.

4. The levels of stress and depression would be negatively associated with obstetric and neonatal outcomes at T3 and these correlations would be mediated by HRV.

## Randomization and allocation

The participants' recruitment will proceed on a continual basis so once begun, a random allocation process will be performed weekly. Online interview will be conducted to ensure eligibility. We use computer-based random number generator to assign numbers to the eligible participants. The research assistant will toss the coin to decide the group allocation and ensure allocation concealment. Two research assistants will interview the participants to check

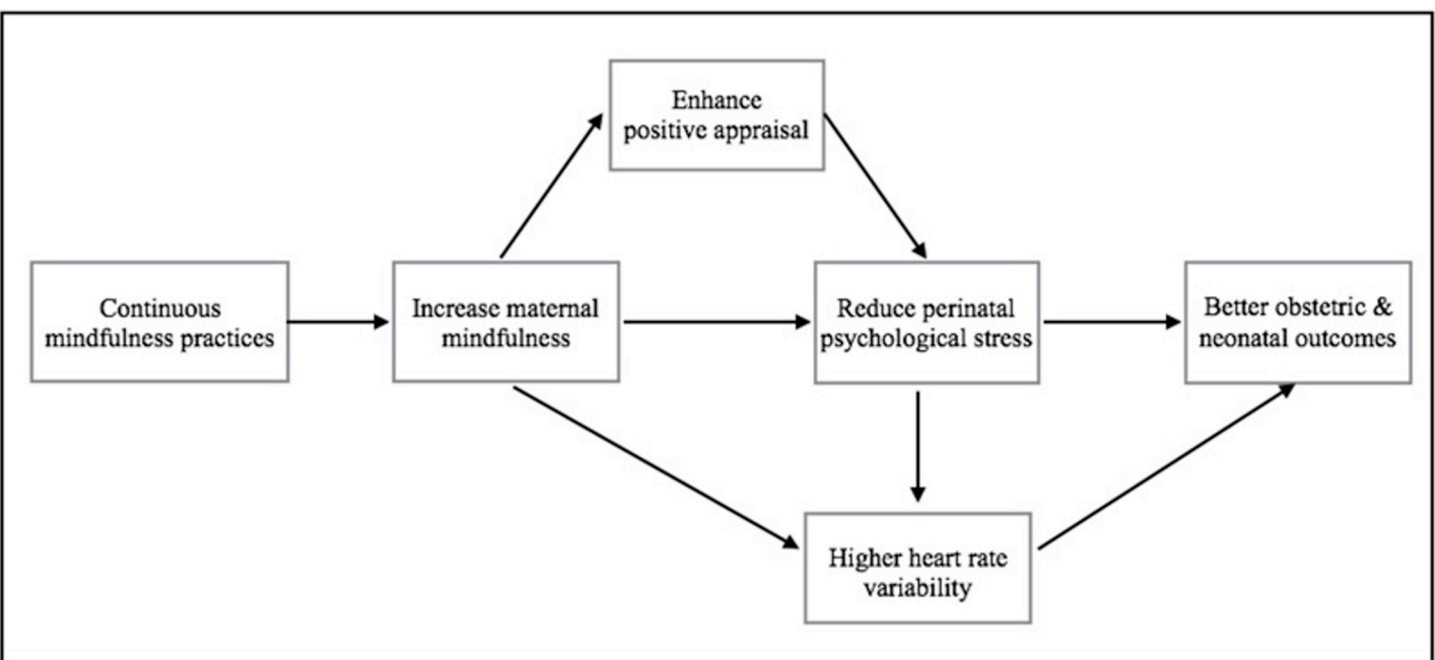

**Fig 1. Conceptual framework.**

eligibility and obtaining consent. One of the two research assistants will assign random number to the participants, and allocate participants to intervention or control group.

## Blinding

It is not possible to blind the participants to group allocation for psychological interventions; however, the study purpose will be masked to participants. The study statistician will also be blinded to the group allocation. Participants' personal identifiers will not be included in the dataset.

## Intervention

Participants in the experimental group will receive the GMPMI. A new participant (a pregnant woman in her second trimester) is expected to complete the eight EBMI lessons in eight weeks and practice mindfulness for about 30–60 minutes daily. The eight-session mindfulness-based intervention is developed from the Mindfulness-Based Cognitive Behaviour (MBCT) protocol for pregnant women. It integrated four-immeasurable meditation, including loving-kindness, compassion, appreciative joy, and equanimity meditations. Different themes are introduced to the participants weekly during the intervention period. These themes include 'deepening capacity for present', 'connecting with breath and body', 'inner inspection', 'raise awareness of your thoughts', and 'savouring goodness'. Prompts and guidance for daily mindfulness practice will be sent to each participant through a social media platform. Participants will also be asked to keep a log of their daily mindfulness practice from T0 to T3 using Google Form. An assisting therapist will be available for online support and will initiate a chat every week throughout the intervention period. The chats will focus on participants' experiences of or difficulties in the mindfulness practice. The assisting therapist will be supervised by an experienced mental health practitioner and a mindfulness teacher.

## Control condition

To control for attention and placebo effects, each new participant in the control group will receive a weekly web-based psychoeducation program for perinatal care (WPPP). WPPP is the parallel program this study adopted for the control group. In this program, essential perinatal and postnatal care knowledge are provided to the participants. Neither mindfulness elements nor counseling support will be given. An assisting therapist will also be available for online support and will initiate a chat every week throughout the intervention period. The chats will focus on participants' experiences of the psychoeducation program.

## Measures

The outcome measures of the study are summarized in Table 1, including the detailed time-points at which the measures will be taken. In order to fulfill the study objective, the primary outcome variables include a battery of scales to measure MPS, mindfulness, positive appraisal and HRV. Secondary outcome variables include more general measures of psychological and physical wellbeing. Anxiety is considered as a more general measure which could be influenced by life events other than pregnancy, and is thus included as a secondary measure. To investigate the impact of the intervention on children, clinical obstetric and neonatal outcomes will also be collected. Other individual data such as demographic data and social support levels will be measured. All the questionnaire links will be sent to the participants through WhatsApp at each time-point for them to fill in on their own devices.

**Table 1. Outcome variables.**

| Variables | Measures | Timepoints |
|---|---|---|
| **Primary outcome variables** | | |
| | Maternal psychological stress | |
| General stress | Perceived Stress Scale –10 items–The scale measures subjective perception of stress with two subscales, perceived helplessness and perceived self-efficacy [36, 37]. | T0, T1, T2, T3, |
| Pregnancy- specific stress | Prenatal Distress Questionnaire– 12 items–The scale consists of three factors–concerns about birth and baby, weight and body image, emotions and relationships [38]. | T0, T1, T2 |
| Depression | Edinburgh Postnatal Depression Scale–Chinese– 10 items–The scale has been validated in the Chinese population with satisfactory psychometric properties [39]. | T0, T1, T2, T3 |
| | Mindfulness | |
| State mindfulness | Short-form Five Facet Mindfulness Questionnaire– 20 items–The scale consists of five subscales–observing, describing, acting awareness, non-judging of inner experience and non-reacting to inner experience [40]. | T0, T1, T2, T3 |
| Daily mindfulness | Daily Mindful Responding Scale– 4 items–A measure designed to assess mindful responding in the daily lives of people undergoing mindfulness training [41]. | Every week until T3 |
| | Positive appraisal | |
| Coping | Prenatal Coping Inventory– 22 items–consists of four subscales–preparation, avoidance, positive appraisal and prayer [42]. | T0, T1, T2 |
| | Heart rate variability | |
| HRV Index | CorSense by Elite HRV–A merchandise device monitoring heart rate variability | T0, T1, T2, T3 |
| **Secondary outcome variables** | | |
| | Psychological wellbeing | |
| Anxiety | Short-form State subscale of the State-Trait Anxiety Inventory– 6 items—The scale has been validated in pregnant women with satisfactory psychometric properties [43]. | T0, T1, T2, T3, |
| Affect | Positive & Negative Affect Subscales of Body-Mind-Spirit Wellbeing Inventory– 8 and 16 items—The two subscales measure common positive and negative affect in Chinese adults [44]. | T0, T1, T2, T3, |
| Spirituality | Chinese Daily Spiritual Experience Scale– 16 items–The scale measures a person's perception of transcendence involvement in daily life [45, 46]. | T0, T1, T2, T3, |
| | Physical wellbeing | |
| Stagnation | Stagnation Scale– 16 items—The scale measures a cluster of mind and body obstruction-like symptoms depicted in Chinese medicine. It has three factors: body/mind obstruction, affect/posture inhibition, and overattachment [47, 48]. Stagnation was identified as a mediator in explaining the mechanism of change in mindfulness [49]. | T0, T1, T2, T3, |
| **Clinical outcomes** | | |
| Obstetric outcomes | Gestational age at birth, pregnancy complications, mode of birth. | T3 |
| Neonatal outcomes | Birth weight, head circumference, months of maturity, Apgar score. | T3 |
| **Individual factors** | | |
| Socio-demographic | Age, gender, marital status, family income, employment, education, gestational age, parity, obstetric history, medical history, pre-pregnancy BMI. | T0 |
| Social support | Prenatal Social Support– 4 items–The scale measures the level of social support women receive during pregnancy [50]. | T0, T1, T2 |

HRV index will be collected through CorSense device. This device is designed as a finger clip which is portable and easy for the pregnant women to use. HRV scores will be calculated automatically. The higher HRV score indicates a better mental health status. The calculation of HRV score is based on the components of the HRV measurements. The R-R intervals will be captured through the Corsense monitor, followed by an industry standard RMSSD calculation. After applying a natural log (ln) to RMSSD, HRV score is conceptualized with a magnitude ranging from 0 to 100. In the meantime, the Corsense device can also capture the raw data of standard deviation of NN intervals (SSDN), the root mean square of the successive difference (rMSSD) in R-R intervals, and low frequency (LF), high frequency (HF), and very low frequency (VLF). This can serve as complementary data for the research team to do data analysis. Elite HRV provides a Team Dashboard for the research team to manage the participants' data in a more convenient way. All the HRV scores and raw data will be sent to the Team Dashboard once the participant completed the measurement.

## Data management plan

HYL and AW will be responsible for data collection under the supervision of SMN, the principal investigator of this research project. All digital data, including questionnaires, consent forms and HRV index data will be stored as a data package in an encrypted hard drive, strictly following the data management guidelines of the University of Hong Kong. Identifiable personal data will be stored separately from the anonymous data to avoid identification of personal data during handling. Participants' confidentiality will be protected by following the ethics guidelines of the Human Research Ethic Committee of the University of Hong Kong.

Data will be analyzed by HYL and AW after the completion of all stages of data collection without interim data analysis. Data will not be shared before the completion of the research project. Data will be available upon request to the corresponding author after the completion of the study.

## Safety considerations

To our knowledge, there are no safety issues for pregnant women in practicing mindfulness or receiving maternal and neonatal care education. A safety exercise review will be provided for participants during the intake interview and will include the local GP contact, project staff contact and suggestions of a safe place to practice. The participants will be encouraged to report any adverse events occurring during practice as soon as possible by WhatsApp. We also have an experienced obstetrics and gynaecology specialist to provide advice to the participants as required to ensure their safety. Any adverse events will be reported by the principal investigator to the Human Research Ethics Committee at the University of Hong Kong using an adverse event report.

## Ethical consideration and declarations

This trial was approved by the Human Research Ethics Committee of the University of Hong Kong (HREC Reference No.: EA1812034). Participants assigned to the intervention group will be voluntarily participate in the current trial. Participants assigned to the control group will receive the psychoeducation program (treatment as usual). Online written informed consent will be provided to the participants in both intervention and control groups after being informed in writing about the study. Participants from both intervention and control groups will fill out questionnaires as listed in the measure part of the article.

## Data analysis

Quantitative data including scale measures and the HRV index will be collected at four time-points as articulated above.

Intention-to-treat analysis will be performed and missing data due to drop–out will be handled by the 'multiple imputation' technique [51]. 1) Using a two-arm RCT design, the efficacy of the intervention will be examined by repeated measures MANOVA on both the primary and secondary outcome variables, while adjusting for individual factors. Partial eta-squared values will be calculated to measure the effect size; values of .02, .13 and .26 suggest small, medium, and large effect sizes, respectively [52]. 2) To investigate the within-group effect, serial trend analysis from T0 to T3 will be conducted on the primary and secondary outcome variables. 3) A series of linear mixed-effects models will be fitted to assess the mediation effects. 4) The software Elite HRV Team Dashboard will be used for HRV data collection. Three HRV indexes will be computed for analysis, namely RMSSD (Mean squared difference between

consecutive normal-to-normal intervals), SDNN (standard deviation of beat-to-beat intervals) and HF (absolute power of HF band).

## Data monitoring

Considering the timeframe and the minimal risk known to the participants of the current study, the Data Monitoring Committee was not formed [53]. The research team will continue monitoring the needs of forming such team.

## Strategies for improving adherence

Participants will be requested to take an HRV measure after everyday mindfulness practice. The measurements will be collected through a computer software Elite HRV Team Dashboard, allowing the research assistant (RA) to track the adherence of each participant. For those participants who failed to practice mindfulness, the RA will send a reminder through WhatsApp.

## Trial status and timeline

The tentative schedule of the research is shown in the Figs 2 and S1. As of 15 April 2021, the study team is preparing the research materials, with recruitment expected to start in May. Intervention and data collection is expected to be completed by May 2022. Data analysis and report writing will be completed by the end of December 2022. Any amendment or deviation from the protocol will be reported to the Human Research Ethics Committee at the University of Hong Kong by submitting an application for amendment of an approved project form in hardcopy.

## Discussion

The study's objective is to test the efficacy of GMBPMI in improving mindfulness and positive appraisal, and reducing pregnant related stress and depression. As a possible mediator, HRV will also be examined to confirm the efficacy of GMBPMI on ANS changed among pregnant women. Additionally, trajectory of maternal psychological distress will also be examined in relation to obstetric and neonatal outcomes at 5-week postpartum period. In line with the research objectives, the primary outcomes of the research are changes in maternal psychological distress, mindfulness, positive appraisal and the HRV index. The secondary outcomes are general psychological and physical wellbeing, such as anxiety, affect, spirituality and stagnation. The clinical outcomes include obstetric and neonatal outcomes, such as gestational age at birth, pregnancy complications, mode of birth, infant Apgar score and head circumference.

This study has both strengths and limitations. Strengths include the randomized controlled trial study design with a large sample size of 198 to ensure sufficient power to demonstrate the efficacy of GMBPMI intervention. More importantly, the research stands out in its innovative delivery mode. Delivered through mobile devices, this study is an extension of the previous face-to-face EMBI intervention, making it feasible for pregnant women to practice at home. Furthermore, this study is one of the few research projects to examine not only psychological but also physiological changes (e.g., HRV), allowing us to detect the potential mediation effect of HRV on GMBPMI and psychological wellbeing. A limitation of this study is that participants in the control group might seek other services or interventions similar to MBI to improve their psychological wellbeing, potentially leading to difficulties in detecting the group difference. To deal with such contamination, we will collect data of any alternative treatments the participants are taking.

| | STUDY PERIOD | | | | |
|---|---|---|---|---|---|
| | Enrolment | Allocation | Post-allocation | | Close-out |
| **TIMEPOINT** | *-t₁* | **0** | *t₁* | *t₂* | *t₃* |
| **ENROLMENT:** | | | | | |
| **Eligibility screening** | X | | | | |
| **Allocation** | | X | | | |
| **Informed consent** | | X | | | |
| **INTERVENTIONS:** | | | | | |
| *[Perinatal Mindfulness Intervention]* | | ●━━━━━━● | | | |
| *[Perinatal Psychoeducation Programme]* | | ●━━━━━━● | | | |
| **ASSESSMENTS:** | | | | | |
| *[Demographic data]* | | X | | | |
| *[Primary outcome variables]* | | X | X | X | X |
| *[Secondary outcome variables]* | | X | X | X | X |
| *[Clinical outcomes]* | | | | | X |
| *[Social Support]* | | X | X | X | |

**Fig 2. SPIRIT schedule of enrolment, intervention and assessments.** Note: -t1, 0, baseline; t1, post 8-week intervention; t2, 36-week gestation follow up; t3. 5-week postpartum follow up.

A significant part of the current study is that the results of the research (reducing maternal psychological distress by GMBPMI) are to be disseminated not only in Hong Kong SAR but also internationally. Hence, we propose the following approach: Upon completion of the study, our research team will seek to publish the data output in international peer reviewed scientific journals, as well as disseminating the results through presentations at local and international conferences. The research material will be further refined by reflecting on the obstacles during the intervention procedure. A guideline booklet will be generated for the general public to use. Furthermore, with the success of this research, it is expected that the whole program will be promoted through the hospital authority to private clinics and public hospitals as supportive resources for pregnant women. All the materials are expected to be accessed through

online platforms either on the Department of Health website by the government of the Hong Kong Special Administrative Region Family Health Service—Home (fhs.gov.hk), or through a privately created webpage by the research team.

## Supporting information

**S1 Fig. CONSORT diagram.**
(TIFF)

**S1 File.**
(DOCX)

**S2 File. SPIRIT checklist.**
(PDF)

**S3 File. All items from the World Health Organization Trial Registration Data Set.**
(PDF)

## Acknowledgments

The authors are grateful to the clinical sites for their contributions to the implementation of the study.

## Author Contributions

**Conceptualization:** Siu-man Ng, Ka Po Chan, Shuang Lu.

**Data curation:** Amenda Wang, Hui Yun Li.

**Funding acquisition:** Siu-man Ng, Ka Po Chan, Hay-ming Herman Lo, Albert Yeung, Shuang Lu.

**Investigation:** Siu-man Ng, Ka Po Chan.

**Methodology:** Siu-man Ng, Ling Li Leng, Ka Po Chan.

**Project administration:** Siu-man Ng, Ling Li Leng, Amenda Wang, Hui Yun Li.

**Resources:** Ling Li Leng, Ka Po Chan.

**Supervision:** Siu-man Ng, Ka Po Chan, Hay-ming Herman Lo, Albert Yeung, Shuang Lu.

**Writing – original draft:** Siu-man Ng, Ling Li Leng, Hui Yun Li.

**Writing – review & editing:** Siu-man Ng, Ling Li Leng, Ka Po Chan, Hay-ming Herman Lo, Albert Yeung, Shuang Lu, Amenda Wang, Hui Yun Li.

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
