## [Decision Letter · Decision Letter 0]

17 Feb 2022

PONE-D-21-16157Study protocol of guided mobile-based perinatal mindfulness intervention (GMBPMI) - a randomized controlled trialPLOS ONE

Dear Dr. Li,

Thank you for submitting your manuscript to PLOS ONE. After careful consideration, we feel that it has merit but does not fully meet PLOS ONE’s publication criteria as it currently stands. Therefore, we invite you to submit a revised version of the manuscript that addresses the points raised during the review process.

The manuscript has been evaluated by two reviewers, and their comments are available below.

The reviewers have raised a number of major concerns regarding statistical analyses, study methodology and more. Could you please carefully revise the manuscript to address all comments raised?

We look forward to receiving your revised manuscript.

Kind regards,

Sebastian Shepherd

Associate Editor

PLOS ONE

Journal Requirements:

2. Thank you for stating the following financial disclosure: "This research received General Research Fund funding scheme (17603520) from Research Grants Council, Hong Kong SAR. SMG is the principal investigator of this research project. The funders had and will not have a role in study design, data collection and analysis, decision to publish, or preparation of the manuscript. "

We note that one or more of the authors is affiliated with the funding organization, indicating the funder may have had some role in the design, data collection, analysis or preparation of your manuscript for publication; in other words, the funder played an indirect role through the participation of the co-authors. If the funding organization did not play a role in the study design, data collection and analysis, decision to publish, or preparation of the manuscript and only provided financial support in the form of authors' salaries and/or research materials, please do the following:

a. Review your statements relating to the author contributions, and ensure you have specifically and accurately indicated the role(s) that these authors had in your study. These amendments should be made in the online form.

b. Confirm in your cover letter that you agree with the following statement, and we will change the online submission form on your behalf: 

“The funder provided support in the form of salaries for authors [insert relevant initials], but did not have any additional role in the study design, data collection and analysis, decision to publish, or preparation of the manuscript. The specific roles of these authors are articulated in the ‘author contributions’ section.

4. Please upload a copy of Figure 2, to which you refer in your text on page 11. If the figure is no longer to be included as part of the submission please remove all reference to it within the text.

Reviewers' comments:

Reviewer's Responses to Questions

**Comments to the Author**

1. Does the manuscript provide a valid rationale for the proposed study, with clearly identified and justified research questions?

Reviewer #1: Partly

Reviewer #2: Partly

2. Is the protocol technically sound and planned in a manner that will lead to a meaningful outcome and allow testing the stated hypotheses?

Reviewer #1: Partly

Reviewer #2: Partly

3. Is the methodology feasible and described in sufficient detail to allow the work to be replicable?

Reviewer #1: Yes

Reviewer #2: Yes

4. Have the authors described where all data underlying the findings will be made available when the study is complete?

Reviewer #1: Yes

Reviewer #2: Yes

5. Is the manuscript presented in an intelligible fashion and written in standard English?

Reviewer #1: Yes

Reviewer #2: Yes

6. Review Comments to the Author

You may also provide optional suggestions and comments to authors that they might find helpful in planning their study.

Reviewer #1: Important note: This review pertains only to ‘statistical aspects’ of the study and so ‘clinical aspects’ [like medical importance, relevance of the study, ‘clinical significance and implication(s)’ of the whole study, etc.] are to be evaluated [should be assessed] separately/independently. Further please note that any ‘statistical review’ is generally done under the assumption that (such) study specific methodological [as well as execution] issues are perfectly taken care of by the investigator(s). This review is not an exception to that and so does not cover clinical aspects {however, seldom comments are made only if those issues are intimately / scientifically related & intermingle with ‘statistical aspects’ of the study}. Agreed that ‘statistical methods’ are used as just tools here, however, they are vital part of methodology [and so should be given due importance]. To improve the article/presentation, clues/hints may be taken from this review but should not limit the process by adhering to those points alone.

COMMENTS: Please refer to section on ‘Sample size calculation’ where it is said that “The previous eastern-based meditation intervention (EBMI) study reported an effect size of 0.44 in reducing prenatal distress” but a reference is not quoted. Therefore, which previous [eastern-based meditation intervention (EBMI)] study authors are referring to is not clear. This is important because the ‘effect size’ of 0.44 [in reducing prenatal distress] used for required ‘Sample size’ calculations here [even overall moderate effect size of 0.4] is too large in my opinion as in such studies ‘effect size’ achieved is generally small and therefore, you need a large sample. Moreover, since the entire study is ‘via telephone or audio-chat / web-based’, “power achieved” is likely to be smaller than intended.

Agreed that ‘Blinding’ is not possible for psychological interventions; however, I wonder if ‘waitlisting controls’ be useful {to some extent}? In addition, please note the following:

Though the measures/tools used are appropriate [Table 1. Outcome variables], most of them yield data that are in [at the most] ‘ordinal’ level of measurement [and not in ratio level of measurement for sure {as the score two times higher does not indicate presence of that parameter/phenomenon as double (for example, a Visual Analogue Scales VAS score or say ‘depression’ score)}]. Then application of suitable non-parametric test(s) is/are indicated/advisable [even if distribution may be ‘Gaussian’ (i.e. normal)]. Agreed that there is/are no non-parametric test(s)/technique(s) available to be used as alternative in all situation(s) [suitable / most desired/applicable], but should be used whenever/wherever they are available.

This is not to say that what is proposed to be used here [Analyses will follow an intention-to-treat method and repeated measures MANOVA will be conducted to compare changes in primary and secondary outcomes] is not correct. In fact, this is most indicated. Since this is a ‘Study protocol’, note this fact in addition to think of ‘change score(s)’. Also note that ‘repeated measures MANOVA’ need/s to be followed by ‘multiple comparison(s)’ with appropriate adjustment in P-value(s).

Is the imputation method proposed [missing data due to drop–out will be handled by the ‘last observations carried forward’ principle] is suitable for this situation? Since missing data are to be imputed using the ‘last observation carried forward’, I hope the authors are aware of disadvantages [like this method assumes that the response remains constant at the last observed value. This assumption can be biased if the timing and the rate of withdrawal is related to the treatment (e.g. in the case of degenerative diseases, using the last observed value to impute for missing data at a later point in the study means that a higher observation will be carried forward, resulting in an overestimation of the true end-of study measurement)] of the method (must be known to these learned authors, still may please be noted) that:

“according to available literature [example, “Inference and Missing Data,” Biometrika, 1976, vol:63, 581–592 and “Multiple Imputation After 18+ Years,” Journal of the American Statistical Association, 1996, vol:91, 473–489] ‘Multiple Imputation’ technique is preferred [considering MCAR (Missing Completely At Random) expected nature of data] than {despite being time-consuming and involving much more computations} compare to all out of other important imputation techniques frequently used [like Group Means, Hot-deck Imputation, Baseline Observation Carried Forward (BOCF), Worst Observation Carried Forward (WOCF), Predicted Mean, and even Last Observation Carried Forward (LOCF)].”

Except these few points, this protocol manuscript appears to be alright.

Reviewer #2: The manuscript is about a study protocol of guided mobile-based perinatal mindfulness intervention (GMBPMI) for pregnant women. But the manuscript doesn’t follow the checklist of SPIRIT 2013 strictly which defining standard protocol items for clinical trials.

1."Maternal psychological distress (MPS) " needs to be defined. This not a term in common use.

2.Why the author recruited pregnant women in their second trimester as participants? Does the pregnant women who attending the study complicate with maternal psychological distress?

3.What the differences between mobile-based and web based? (From the sentence: Eligible participants will be randomized to receive either the guided mobile-based perinatal mindfulness intervention (GMPMI) or the webbased perinatal psychoeducation program (WPPP)).

4. Why the participants in control need to receive web-based psychoeducation program?

What kinds of strategies for psychological counseling will be used for women in control group?

5.Where was the study being conducted? The author did not describe study setting in detaile.

6. How do the author generate the allocation sequence? Is there any measure to conduct the allocation concealment?

7. Who will generate the allocation sequence, who will enroll participants, and who will assign participants to interventions?

8. Why anxiety was defined as the second outcome rather than the first outcome?

9. How to evaluate HRV? What is the difference between CorSense by Elite HRV and KardaMobile ECG ?

10. What is the content for eight weekly interventions? Is there any difference among each session?

7. PLOS authors have the option to publish the peer review history of their article (what does this mean?). If published, this will include your full peer review and any attached files.

Reviewer #1: No

Reviewer #2: No

---

## [Author Response · Author response to Decision Letter 0]

7 Mar 2022

Dear Dr. Shepherd and Reviewers,

 We are very glad to receive a positive editorial decision and we appreciated your time to provide us with thoughtful and insightful feedbacks and comments on this study protocol. After thoroughly reviewing your feedbacks and comments, we would like to take this opportunity to further improve our manuscript. Some misunderstandings which might be due to the poor presentation of the manuscript will also be clarified. The following shows the point-to-point response to both reviewers’ comments and questions. Full text of each editor’s and reviewer’s comments are copied as below. Each point will be followed with a reply immediately. The manuscript is also carefully revised based on the reviewers’ comments. Files of 'Response to Reviewers', 'Manuscript', 'Revised Manuscript with Track Changes' are attached to this submission for your further review.

Journal Requirements

1. Please ensure that your manuscript meets PLOS ONE's style requirements, 

including those for file naming. The PLOS ONE style templates can be found at 

Reply

The manuscript and file names are edited to meet PLOS One’s style requirements. Please refer to the newly submitted manuscripts.

2. Thank you for stating the following financial disclosure: "This research received 

General Research Fund funding scheme (17603520) from Research Grants Council, Hong Kong SAR. SMG is the principal investigator of this research project. The funders had and will not have a role in study design, data collection and analysis, decision to publish, or preparation of the manuscript. "

We note that one or more of the authors is affiliated with the funding organization, indicating the funder may have had some role in the design, data collection, analysis or preparation of your manuscript for publication; in other words, the funder played an indirect role through the participation of the co-authors. If the funding organization did not play a role in the study design, data collection and analysis, decision to publish, or preparation of the manuscript and only provided financial support in the form of authors' salaries and/or research materials, please do the following:

a. Review your statements relating to the author contributions, and ensure you have specifically and accurately indicated the role(s) that these authors had in your study. These amendments should be made in the online form.

b. Confirm in your cover letter that you agree with the following statement, and we will change the online submission form on your behalf: 

“The funder provided support in the form of salaries for authors [insert relevant initials], but did not have any additional role in the study design, data collection and analysis, decision to publish, or preparation of the manuscript. The specific roles of these authors are articulated in the ‘author contributions’ section.

Reply

The funding organization did not play a role in the study design, data collection and analysis, decision to publish or preparation manuscript. Only financial support was provided in forms of author Amenda Wang’s salary and research materials. The statement in section b is included in the cover letter. 

3. Your ethics statement should only appear in the Methods section of your 

manuscript. If your ethics statement is written in any section besides the Methods, please delete it from any other section. 

Reply

The ethics statement section is moved to the Methods section. 

4. Please upload a copy of Figure 2, to which you refer in your text on page 11. If the 

figure is no longer to be included as part of the submission please remove all reference to it within the text.

Reply

The Figure 2 is changed to S2 Fig both in the manuscript and under the Supporting documents section. Both S1 and S2 Figs will be uploaded to the online platform. 

Reviewer #1

Important note: This review pertains only to ‘statistical aspects’ of the study and so ‘clinical aspects’ [like medical importance, relevance of the study, ‘clinical significance and implication(s)’ of the whole study, etc.] are to be evaluated [should be assessed] separately/independently. Further please note that any ‘statistical review’ is generally done under the assumption that (such) study specific methodological [as well as execution] issues are perfectly taken care of by the investigator(s). This review is not an exception to that and so does not cover clinical aspects {however, seldom comments are made only if those issues are intimately / scientifically related & intermingle with ‘statistical aspects’ of the study}. Agreed that ‘statistical methods’ are used as just tools here, however, they are vital part of methodology [and so should be given due importance]. To improve the article/presentation, clues/hints may be taken from this review but should not limit the process by adhering to those points alone.

COMMENTS: Please refer to section on ‘Sample size calculation’ where it is said that “The previous eastern-based meditation intervention (EBMI) study reported an effect size of 0.44 in reducing prenatal distress” but a reference is not quoted. Therefore, which previous [eastern-based meditation intervention (EBMI)] study authors are referring to is not clear. This is important because the ‘effect size’ of 0.44 [in reducing prenatal distress] used for required ‘Sample size’ calculations here [even overall moderate effect size of 0.4] is too large in my opinion as in such studies ‘effect size’ achieved is generally small and therefore, you need a large sample. Moreover, since the entire study is ‘via telephone or audio-chat / web-based’, “power achieved” is likely to be smaller than intended.

Reply

Thank you for the comments and sharing your opinion about statistical power in conducting web-based intervention. The reference regarding EBMI study [28] is now quoted in the manuscript (also listed below for your easy reference). The suggestion of adopting a larger sample is taken. The target sample size of 198 is believed to be conservative enough. We will consider expanding the sample size to ensure enough statistical power after preliminary data analysis. 

28. Chan KP. Effects of perinatal meditation on pregnant Chinese women in Hong Kong: a randomized controlled trial. Journal of Nursing Education and Practice. 2015; 5(1): 1.

Agreed that ‘Blinding’ is not possible for psychological interventions; however, I wonder if ‘waitlisting controls’ be useful {to some extent}? 

Reply

Waitlisting control is not feasible in the current study because the targeted participants are in their second trimester of pregnancy. Upon completion of intervention and follow-up measures in the experimental group, participants in the waitlist control group should have already delivered their baby. 

In addition, please note the following:

Though the measures/tools used are appropriate [Table 1. Outcome variables], most of them yield data that are in [at the most] ‘ordinal’ level of measurement [and not in ratio level of measurement for sure {as the score two times higher does not indicate presence of that parameter/phenomenon as double (for example, a Visual Analogue Scales VAS score or say ‘depression’ score)}]. Then application of suitable non-parametric test(s) is/are indicated/advisable [even if distribution may be ‘Gaussian’ (i.e. normal)]. Agreed that there is/are no non-parametric test(s)/technique(s) available to be used as alternative in all situation(s) [suitable / most desired/applicable], but should be used whenever/wherever they are available.

Reply

Agreed that non-parametric test is worth considering whenever it is needed. We will take this point into consideration while conducting data analysis. 

This is not to say that what is proposed to be used here [Analyses will follow an intention-to-treat method and repeated measures MANOVA will be conducted to compare changes in primary and secondary outcomes] is not correct. In fact, this is most indicated. Since this is a ‘Study protocol’, note this fact in addition to think of ‘change score(s)’. Also note that ‘repeated measures MANOVA’ need/s to be followed by ‘multiple comparison(s)’ with appropriate adjustment in P-value(s).

Is the imputation method proposed [missing data due to drop–out will be handled by the ‘last observations carried forward’ principle] is suitable for this situation? Since missing data are to be imputed using the ‘last observation carried forward’, I hope the authors are aware of disadvantages [like this method assumes that the response remains constant at the last observed value. This assumption can be biased if the timing and the rate of withdrawal is related to the treatment (e.g. in the case of degenerative diseases, using the last observed value to impute for missing data at a later point in the study means that a higher observation will be carried forward, resulting in an overestimation of the true end-of study measurement)] of the method (must be known to these learned authors, still may please be noted) that:

“according to available literature [example, “Inference and Missing Data,” Biometrika, 1976, vol:63, 581–592 and “Multiple Imputation After 18+ Years,” Journal of the American Statistical Association, 1996, vol:91, 473–489] ‘Multiple Imputation’ technique is preferred [considering MCAR (Missing Completely At Random) expected nature of data] than {despite being time-consuming and involving much more computations} compare to all out of other important imputation techniques frequently used [like Group Means, Hot-deck Imputation, Baseline Observation Carried Forward (BOCF), Worst Observation Carried Forward (WOCF), Predicted Mean, and even Last Observation Carried Forward (LOCF)].”

Reply

Thank you for your mindful suggestion. Apart from LOCF, we will also use Multiple Imputation/ Hot-deck Imputation to treat missing data, depending on the characteristics of our missing data. 

Except these few points, this protocol manuscript appears to be alright.

Reviewer #2

The manuscript is about a study protocol of guided mobile-based perinatal mindfulness intervention (GMBPMI) for pregnant women. But the manuscript doesn’t follow the checklist of SPIRIT 2013 strictly which defining standard protocol items for clinical trials.

1."Maternal psychological distress (MPS) " needs to be defined. This not a term in common use.

Reply

Maternal psychological stress refers to the subjective feelings of distress or strain experienced by mothers during their pregnancy. 

2.Why the author recruited pregnant women in their second trimester as participants? Does the pregnant women who attending the study complicate with maternal psychological distress?

Reply

During the second trimester, women in pregnancy are in stable period, and have the capacity to participate in the intervention program. Previous studies have suggested that intervention in the second trimester might be useful to prevent postpartum depression. Hence, we recruited pregnant women in their second trimester as our participants. 

We did not include maternal psychological distress as part of our inclusion and exclusion criteria. Previous research suggested maternal psychological distress is commonly observed among pregnant women in their second and third trimester. Our research aims to investigate the prevention effectiveness of GMBPMI on the maternal psychological distress. 

3.What the differences between mobile-based and web based? (From the sentence: Eligible participants will be randomized to receive either the guided mobile-based perinatal mindfulness intervention (GMPMI) or the webbased perinatal psychoeducation program (WPPP)).

Reply

The main difference between GMPMI and WPPP is the content of the program. GMPMI is a mindfulness-based intervention delivered through mobile phone by using WeChat mini program, and thus it is described as a mobile-based perinatal mindfulness intervention. WPPP is the parallel program this study adopted for the control group. In this program, only psychoeducation content is used, without any mindfulness practice elements. WPPP program is uploaded to an online platform called Youku, where the participants will watch the video clips produced by our team. That’s why we name this program a web-based perinatal psychoeducation program. Since psychoeducation is something basic and essential to women in pregnancy, WPPP will also be provided to the intervention group participants. 

4. Why the participants in control need to receive web-based psychoeducation program? What kinds of strategies for psychological counseling will be used for women in control group?

Reply

The psychoeducation program includes essential maternal and perinatal care knowledge. It is a general education for the pregnant women to take care of themselves and their newborn babies. The program does not include psychological counselling elements.

Another reason to provide WPPP to control group is to ensure motivation for our participants to join the research study. From WPPP they will receive information, which is also beneficial but distinguished from GMPMI program. 

5.Where was the study being conducted? The author did not describe study setting in detail.

Reply

The study team is based in Hong Kong, and uses the local dialect in delivering the intervention. Practically participants will speak local Hong Kong dialect, and be mostly residing in Hong Kong. 

6. How do the author generate the allocation sequence? Is there any measure to conduct the allocation concealment?

Reply

After checking eligibility and obtaining consent from the participants through online interview, we use computer-based random number generator to assign each participant a number. The research assistant will toss the coin to decide the group allocation before the study. The research assistant will not know the group allocation before the study. This assures allocation concealment. 

7. Who will generate the allocation sequence, who will enroll participants, and who will assign participants to interventions?

Two research assistants will interview the participants to check eligibility and obtain informed consent. One of the two research assistants will assign random number to the participants, and assign participants to intervention or control group. 

8. Why anxiety was defined as the second outcome rather than the first outcome?

Reply

In this study, our focus is on the pregnancy-related distress. The primary outcomes include stress and depressive symptoms which are stressor (pregnant) related. Anxiety is more general, including persistent, excessive worries without an obvious stressor. Anxiety could be influenced by other adverse life events other than pregnancy. Hence, we put anciety as a secondary outcome.

9. How to evaluate HRV? What is the difference between CorSense by Elite HRV and KardaMobile ECG?

Reply

HRV will be collected by using the CorSense device. This device is designed as a finger clip which is portable and easy for the pregnant women to use. The other reason is that HRV scores will be calculated automatically. The higher HRV score indicates a better mental health status. The calculation of HRV score is based on the components of the HRV measurements. First, the R-R intervals will be captured through the Corsense monitor. Then, RMSSD calculation, as an industry standard is applied. A natural log (ln) is applied to RMSSD to conceptualize its magnitude. Last, the ln(RMSSD) is expanded to general score ranging from 0 to 100. In the meantime, the Corsense device can also capture the raw data of standard deviation of NN intervals (SSDN), the root mean square of the successive difference (rMSSD) in R-R intervals, and low frequency (LF), high frequency (HF), and very low frequency (VLF). This can serve as complementary data for the research team to do data analysis. Moreover, Elite HRV provides a Team Dashboard for the research team to manage the participants’ data. All the HRV scores and raw data will be sent to the Team Dashboard once the participant completed the measurement. 

KardaMobile ECG captures the raw ECG that requires the research team to do the calculation. Since our intervention is delivered online, our participants will need to report their data frequently to the research team. Since there is no such function that the data could be automatically transferred to a computer terminal, it is not feasible for the research team to collect and manage the data remotely. Requesting the participants, who are pregnant, to report and capture the ECG raw data to the research team might reduce their compliance to the study. Bias could also arise from this.

10. What is the content for eight weekly interventions? Is there any difference among each session?

Reply

The eight-session mindfulness-based intervention is developed from the Mindfulness-Based Cognitive Behaviour (MBCT) protocol for pregnant women. It integrated four-immeasurable meditation, including loving-kindness, compassion, appreciative joy, and equanimity meditations. Different themes are introduced to the participants weekly during the intervention period. These themes include ‘deepening capacity for present’, ‘connecting with breath and body’, ‘inner inspection’, ‘raise awareness of your thoughts’, and ‘savouring goodness’. 

 Thank you very much for all your time and considerate comments and feedbacks. We look forward to having further knowledge exchange with you. Should you have further comments, please feel free to contact us. 

Best regards,

Hui Yun Li

erinhy@connect.hku.hk

---

## [Decision Letter · Decision Letter 1]

16 Jun 2022

Study protocol of guided mobile-based perinatal mindfulness intervention (GMBPMI) - a randomized controlled trial

PONE-D-21-16157R1

Dear Dr. Li,

We’re pleased to inform you that your manuscript has been judged scientifically suitable for publication and will be formally accepted for publication once it meets all outstanding technical requirements.

We noticed that there are some language and grammar errors remaining, and we thus request that you fully copyedit the manuscript before submitting the final version.

Kind regards,

Hanna Landenmark

Staff Editor

PLOS ONE

Additional Editor Comments (optional):

Reviewers' comments:

Reviewer's Responses to Questions

**Comments to the Author**

1. Does the manuscript provide a valid rationale for the proposed study, with clearly identified and justified research questions?

Reviewer #1: Yes

2. Is the protocol technically sound and planned in a manner that will lead to a meaningful outcome and allow testing the stated hypotheses?

Reviewer #1: Yes

3. Is the methodology feasible and described in sufficient detail to allow the work to be replicable?

Reviewer #1: Yes

4. Have the authors described where all data underlying the findings will be made available when the study is complete?

Reviewer #1: Yes

5. Is the manuscript presented in an intelligible fashion and written in standard English?

Reviewer #1: Yes

6. Review Comments to the Author

You may also provide optional suggestions and comments to authors that they might find helpful in planning their study.

Reviewer #1: COMMENTS: Since all of the comments made on earlier draft by me (and hopefully by other respected reviewers also) were/are attended positively, I recommend the acceptance because the manuscript now has achieved acceptable level, in my opinion.

7. PLOS authors have the option to publish the peer review history of their article (what does this mean?). If published, this will include your full peer review and any attached files.

Reviewer #1: **Yes: **Dr. Sanjeev Sarmukaddam

---

## [Editor Report · Acceptance letter]

28 Jun 2022

PONE-D-21-16157R1 

Study protocol of guided mobile-based perinatal mindfulness intervention (GMBPMI) - a randomized controlled trial 

Dear Dr. Li:

I'm pleased to inform you that your manuscript has been deemed suitable for publication in PLOS ONE. Congratulations! Your manuscript is now with our production department. 

Kind regards, 

on behalf of

Dr. Hanna Landenmark 

Staff Editor

PLOS ONE